# The Generation of an Artificial ATP Deficit Triggers Antibiotic Production in *Streptomyces lividans*

**DOI:** 10.3390/antibiotics11091157

**Published:** 2022-08-27

**Authors:** Nicolas Seghezzi, Emmanuelle Darbon, Cécile Martel, Michelle David, Clara Lejeune, Catherine Esnault, Marie-Joelle Virolle

**Affiliations:** Institute for Integrative Biology of the Cell (I2BC), University Paris-Saclay, CEA, CNRS, 91198 Gif-sur-Yvette, France

**Keywords:** phosphate limitation, ATP deficit, oxidative phosphorylation, oxidative stress, ATPase, antibiotics

## Abstract

In most *Streptomyces* species, antibiotic production is triggered in a condition of phosphate limitation, a condition that is known to be correlated with a low intracellular ATP content compared to growth in a condition of phosphate proficiency. This observation suggests that a low ATP content might be a direct trigger of antibiotic biosynthesis. In order to test this hypothesis, we introduced into the model strain *Streptomyces lividans*, a functional and a non-functional ATPase cloned into the replicative vector pOSV206 and expressed under the control of the strong ErmE* promoter. The functional ATPase was constituted by the α (AtpA), β (AtpB) and γ (AtpD) sub-units of the native F1 part of the ATP synthase of *S. lividans* that, when separated from the membrane-bound F0 part, bears an ATPase activity. The non-functional ATPase was a mutated version of the latter, bearing a 12 amino acids deletion encompassing the active site of the AtpD sub-unit. *S. lividans* was chosen to test our hypothesis since this strain hardly produces any antibiotics. However, it possesses the same biosynthetic pathways of various specialized metabolites as *S. coelicolor*, a phylogenetically closely related strain that produces these metabolites in abundance. Our results demonstrated that the over-expression of the functional ATPase, but not that of its mutated version, indeed correlated with the production of the bioactive metabolites of the CDA, RED and ACT clusters. These results confirmed the long known and mysterious link existing between a phosphate limitation leading to an ATP deficit and the triggering of antibiotic biosynthesis. Based on this work and the previous published results of our group, we propose an entirely novel conception of the nature of this link.

## 1. Introduction

Gram-positive, filamentous soil bacteria of the *Streptomyces* genus are of great medical and economic importance, since they produce a huge variety of bio-active molecules useful to human health or agriculture such as antibiotics, anti-cancer drugs, pesticides and herbicides [1,2]. The biosynthesis of these molecules usually takes place when growth slows down or stops. It is thought to be triggered by some nutritional limitation [3], a phosphate limitation being one of the most efficient triggers of specialized metabolites biosynthesis [4]. As anticipated, the extracellular phosphate concentration influences the intracellular ATP concentration. Indeed, it was previously demonstrated that *S. lividans* grown in a Pi limited medium (1 mM) has an ATP content approximately 2-fold lower than that of the same strain grown in a Pi proficient medium (5 mM) [5]. ATP is generated either by substrate level phosphorylation during glycolysis or by oxidative phosphorylation. The latter process involves the re-oxidation of reduced co-factors generated by the TCA cycle by the respiratory chain. This oxidation leads to the extrusion of protons and the resulting proton gradient constitutes the driving force for ATP synthesis by the ATP synthase. As ATP is a biological energy source needed for numerous catabolic, anabolic and transport reactions, its intracellular concentration should be carefully maintained in a rather narrow range by complex homeostatic processes.

In order to determine whether a low ATP content constitutes a trigger of antibiotic biosynthesis in *Streptomyces*, we expressed a functional and a non-functional ATPase in *S. lividans*, a weak antibiotic producing strain. Indeed, this strain possesses the same biosynthetic pathways of various specialized metabolites as the phylogenetically closely related and strong antibiotic producer, *S. coelicolor*, but hardly expresses them. Three of these biosynthetic pathways are well characterized, including that directing the biosynthesis of the peptidyl colorless Calcium Dependent Antibiotic (CDA), the red hybrid peptidyl/polyketide antibiotic, undecylprodigiosin (RED) and of the blue polyketide antibiotic, actinorhodin (ACT). The functional ATPase was constituted by the α (AtpA), β (AtpB) and γ (AtpD) sub-units of the F1 part of the ATP synthase of *S. lividans* (Figure 1) that, when separated from the membrane-bound F0 part, bears an ATPase activity. The non-functional ATPase was constituted by a mutated version of the native F1 part of the ATP synthase of *S. lividans* via the deletion of 12 amino acids of the active site of AtpD. Our RT-PCR results indicated that the over-expression of the native F1 part of the ATP synthase, but not that of the non-functional ATPase, was indeed correlated with a slight growth retardation, a lower ATP content, an earlier and/or stronger expression of some genes of the CDA, RED and ACT pathways. This lead to an enhanced production of the corresponding bioactive metabolites as well as to a lower total lipid content. These results confirmed the long known but mysterious link existing between phosphate limitation leading to a lowering of the ATP content and the triggering of antibiotic biosynthesis. The connections between the triggering of antibiotic production in the weak antibiotic producing strain, *S. lividans*, via the over-expression of an ATPase described in this study and the previously described specific metabolic features of *S. coelicolor* that are correlated with its strong ability to produce an antibiotic [5,6,7,8], led us to propose an entirely novel conception of the nature of the link between an ATP deficit, usually linked to a phosphate limitation, and the triggering of antibiotic biosynthesis in *Streptomyces* species.

## 2. Results

### 2.1. Quantification of Growth, ATPase Expression and Intracellular ATP Content in S. lividans/pOSV206_atpAGD and S. lividans/pOSV206_atpAGD^Δ^

*S. lividans* strains carrying the functional and the non-functional ATPase and cloned under the control of the strong ErmE* promoter in the pOSV206 vector, were grown on a solid HT medium. RNA was prepared from 48 h grown cultures of both strains. Oligos that were specifically designed to differentiate the copy of *atpAGD* carried by the plasmid from the genomic copy (Appendix A) were used to determine the transcription level of the first gene of the construct by RT-PCR. The results shown in Appendix A clearly indicate that both constructs were highly transcribed in *S. lividans*.

Subsequently, a growth curve of both strains was established to determine whether the over-expression of *atpAGD* had an impact on bacterial growth. The results shown in Figure 1 indicate that growth of *S. lividans*/pOSV206_*atpAGD* was slightly slower than that of *S. lividans*/pOSV206_*atpAGD*^Δ^, indicating a slight detrimental effect of the over-expression of *atpAGD* on growth.

The ATP content of *S. lividans*/pOSV206_*atpAGD* and *S. lividans*/pOSV206_*atpAGD*^Δ^ was quantified every 12 h from 36 h to 96 h of growth on a HT solid medium. The results shown in Figure 2 indicated that the ATP content was slightly lower, at least at 60 h, 72 h and 84 h, in the strain expressing the functional ATPase than in the strain expressing the non-functional ATPase. However, the difference in the ATP content of the two strains was not huge, most likely because homeostatic mechanisms, aimed at the restoration of the cellular energetic balance, are triggered when the ATP concentration falls below a certain threshold.

### 2.2. Analysis in RT-PCT of the Transcriptional Level of Expression of Genes of the CDA, RED and ACT Clusters in S. lividans/pOSV206_atpAGD and S. lividans/pOSV206_atpAGD^Δ^

In order to determine whether our constructs had an impact on the level of expression of the three most studied biosynthetic pathways of *S. lividans* and *S. coelicolor* that are the CDA, RED and ACT clusters, RNA was prepared from 24 h, 48 h and 72 h cultures of the two strains grown on a solid HT medium. The level of expression of two genes, including at least one regulatory gene, belonging to each of these three clusters was determined by RT-PCR using oligos shown in Appendix A. The level of expression of the major vegetative sigma factor HrdB, whose expression is stable at the three time points, was used as the internal control. The results shown in Figure 3 indicate the expression of genes of these clusters either occurred earlier and/or was stronger in the strain expressing the functional ATPase than in the strain expressing the non-functional ATPase. This indicates that an ATP deficit indeed plays a role in the triggering of the expression of these three pathways.

The level of expression of a single gene of five other pathways was also tested. The results of Appendix A indicate that *cpkA* encoding a PKS of type I involved in the biosynthesis of a cryptic yellow polyketide, *eizA* involved in the biosynthesis of albaflavenone and the NRPS *sco6431* showed a similar trend as the CDA, RED and ACT. In contrast, the level of expression of *crtE* involved in carotenoid biosynthesis and that of *cchH* (NRPS) involved in coelichelin biosynthesis was similar in both strains. This indicates that an ATP deficit had not had the same impact on all of the biosynthetic pathways.

### 2.3. Assay of CDA, RED and ACT Production in S. lividans/pOSV206_atpAGD and S. lividans pOSV206_atpAGD^Δ^ Grown in HT or R2YE

In order to quantify the level of RED production, these strains were grown in a HT medium for 48 h (Figure 4A). To estimate the level of CDA production, these two strains were grown on an Oxoid Nutrient Agar (ONA) medium in absence or presence of calcium, as described in [9] (Figure 4B). The CDA and RED were clearly more abundantly produced by the strain expressing the functional ATPase than by the strain expressing the non-functional ATPase (Figure 4A,B), which is consistent with the RT-PCR data (Figure 3).

Since ACT production was weak in the HT medium, the strains were grown on a solid R2YE medium with no K_2_HPO_4_ added (in this medium, free phosphate coming from elements of the growth medium was at a concentration of 1 mM final, in a condition of phosphate limitation). The results of Figure 5 indicate that the strain over-expressing the functional ATPase was deep blue, indicating ACT production, whereas the strain expressing the non-functional ATPase was not. This is consistent with RT-PCR data obtained in HT medium (Figure 3). ACT was assayed at various points throughout the growth in the two strains. ACT production was below detection limits in the strain expressing the non-functional ATPase, whereas the ACT was clearly detectable in the strain expressing the functional ATPase (Figure 5). ACT production started to be detectable in *S. lividans*/pOSV206_*atpAGD* after 60 h of growth and keep increasing between 72 h and 80 h. Interestingly, most ACT remained intracellular.

### 2.4. Analysis of the Total Lipid Content of the Strains S. lividans/pOSV206_atpAGD and S. lividans/pOSV206_atpAGD^Δ^

Since it was previously demonstrated that there is a negative correlation between total lipid content and antibiotic production in the *ppk* and *pptA* mutants of *S. lividans* [10], in *S. coelicolor* [5] as well as in other *Streptomyces* strains [11], we assessed the lipid content of the two strains grown on solid R2YE with no phosphate added for 72 h. The results shown in Figure 6 confirm that the total lipid content of the strain expressing the functional ATPase was significantly lower than that of the strain expressing the non-functional ATPase. These data confirmed, once again, the existence of a negative correlation between total lipid content and antibiotic production [5,6,11].

## 3. Discussion

In this paper we demonstrated that a decrease in the intracellular ATP level, due to the over-expression of an ATPase, is correlated with the lowering of the total lipids content and the triggering of the biosynthesis of specific bio-active specialized metabolites in the model strain, *S. lividans*, that usually does not produce these metabolites in the conditions used in this study. Based on the results of this study and of other studies published by our group [5,7,10,12], we propose a novel understanding of the link existing between these processes. The over-expression of the ATPase, that somehow mimics and accentuates the effect of a phosphate limitation, would lead to a decrease of the intracellular ATP concentration below a certain threshold. This would trigger the activation of the oxidative metabolism that generates the ATP in order to restore the energetic balance of the cell [5,7,10,12]. Such activation would require the fueling of the TCA with acetyl-coA, that therefore could not be used for lipid biosynthesis resulting into a reduced total lipid content [5,6,7]. The TCA cycle generates reduced co-factors, whose re-oxidation by the respiratory chain yields ATP as well as reactive oxygen or nitrogen species (ROS and NOS) and thus oxidative stress (OS). OS was proposed to be an important trigger of ACT biosynthesis, [6] since ACT was shown to bear anti-oxidant properties [7]. The anti-oxidant function of the ACT was attributed to its ability to capture excess electrons of ROS/NOS thanks to its quinone groups [7]. Furthermore, since the onset of ACT production was shown to coincide with an abrupt drop of the high ATP content of *S. coelicolor* [5], ACT would also have an anti-respiratory function via its ability to capture electrons of the respiratory chain. Interestingly, in the *S. lividans* strain over-expressing the ATPase, ACT remains mainly intracellular. This suggests that the anti-respiratory and anti-oxidant functions of this molecule are required intracellularly to counteract the potential detrimental consequences of the energetic stress imposed by the high and constitutive expression of the functional ATPase. Furthermore, the enhanced production of CDA [13] and RED [14] in this context of energetic stress suggests that these molecules, as ACT, would also have regulatory functions. Indeed, CDA and RED are both known to alter the integrity of the phospholipid membrane. They might thus contribute to the dissipation of the gradient proton, leading to a reduction of respiration efficiency and thus of ATP generation. The function of these three bioactive molecules would be to reduce, by different strategies, the respiration efficiency of the strain, to limit oxidative stress as well as the generation of ATP, especially in the condition of phosphate scarcity. Since CDA and RED biosynthesis always occurs before ACT biosynthesis in *S. coelicolor* [12], the induction of CDA and RED biosynthesis might occur in condition of mild energetic/oxidative stress whereas ACT biosynthesis would occur in the condition of a more severe energetic/oxidative stress. Consistently, these three molecules were shown to be constitutively and abundantly produced in *S. coelicolor* that is characterized by a highly active oxidative metabolism [5,8,15].

## 4. Materials and Methods

### 4.1. Plasmid Construction

In order to avoid the likely introduction of mutations by PCR amplification of a long fragment, a 4.3 kb long *Apa*I fragment carrying the genes *atpA* (*sli_5640*), *atpG* (*sli_5641*) and *atpD* (*sli_5642*) encoding the α, γ and β sub-units of the F1 part of ATP synthase, respectively, was purified from the cosmid 2St6G5 [13]. To do so, *Apa*I restriction sites, one located 24 bp upstream of the *atpA* initiation codon and containing the ribosome binding site and the other 280 bp downstream of the *atpD* gene termination codon, were used. The isolated atpAGD fragment was purified, blunt ended with T4 polymerase and cloned into the pUC19 [14] digested with *Sma*I (Fermentas). This construction was used to make a 36 bp deletion within *atpD*, leading to a deletion of 12 amino acids (residues 163 to 174) comprising the nucleotide binding site of AtpD and thus inactivating the ATPase function of the F1 part of the ATP synthase. To do so, a PCR-directed mutagenesis was used. The mutated and complementary primers 1 (5′-GATCGGTCTGCAGGAGATGATCTACCGCGT-3′) and 2 (3′-TCCCGCCGTTCTAGCCAGACGTCCTCTACT-5′) were used to make the deletion and create a *Pst*I site, allowing the distinction between the native and mutated constructs. Primers mapping on each site of the deleted region were used to amplify the deleted fragment, and unique *Sca*I and *Xmn*I restriction sites were used to replace the native fragment by the deleted one in pUC19 yielding pUC19_*atpAGD* and pUC19_*atpAGD*^Δ^. The native and mutated *atpAGD* fragments cut by *Hind*III and *Eco*RI were cloned into the conjugative/replicative plasmid pOSV206 to give the plasmids pOSV206__*atpAGD* (active native ATPase) and pOSV206_*atpAGD*^Δ^ (inactive mutated ATPase) that were transferred by conjugation from *E. coli* to *S. lividans* as described in [16]. pOSV206 is a derivative of the replicative plasmid pUWL201 [17] in which the thiostrepton resistance gene (*tsr*) [18] was replaced by the apramycin resistance cassette *aac*(3)IV [19] and the origin of replication OriT [16] was added to make it conjugative (Maud Juguet, unpublished results).

### 4.2. Strains and Culture Conditions

The strains used in this study are derived from *S. lividans* TK24, transformed by the replicative and conjugative plasmids pOSV206_*atpAGD* and pOSV206_*atpAGD*^Δ^. *Streptomyces* growth conditions and manipulations were performed following the protocols of the Practical Streptomyces Genetics manual [20]. A Soya Flour Mannitol (SFM) medium was used to prepare the spores of the strains. A solid HT medium, with apramycin (Apr) at 30 μg/mL, was used to establish growth curves in dry biomass (Figure 2A), to prepare RNA for RT-PCR experiments (Figure 3 and Appendix A), as well as to assay intracellular ATP (Figure 2B) and RED production (Figure 4A). Whereas, a R2YE medium limited in phosphate (1 mM, no K_2_HPO_4_ added, with apramycin at 50 μg/mL) [21] was used to assay ACT production (Figure 5) and a ONA medium was used to assay the CDA production [9].

The strain of *E. coli* DH5α was used for sub-cloning and the strain of *E. coli* S17-1 [22] was used for plasmid transfer (conjugation) between *E. coli* and *S. lividans* TK24.

### 4.3. Extraction and Assay of Intracellular ATP Concentrations

The extraction and assay of intracellular ATP concentrations were carried out from three independent replicates as described in [5].

### 4.4. RNA Preparation

RNA was isolated from mycelia of *S. lividans/*pOSV206_*atpAGD* and *S. lividans*/pOSV206_*atpAGD*^Δ^ grown for 24 h, 48 h and 72 h at 28 °C on the solid HT medium. In order to preserve RNA integrity, the collected mycelium was immediately frozen in liquid nitrogen in a solution containing denaturing guanidinium thiocyanate buffer RA1 (Macherey-Nagel, Hoerdt, France), phenol-chloroform and ß-mercaptoethanol (a reducing agent). The cells were then lysed and homogenized in the presence of glass beads (diameter < 106 µm) using a Fast-Prep apparatus (Savant Instruments, Telangana, India). Total The RNA was purified using the Nucleospin RNA Kit (Macherey-Nagel, Hoerdt, France), according to the manufacturer’s instructions. To remove residual DNA, a DNase TURBO™ treatment (Invitrogen, Illkirch, France) was performed at 37 °C for 1 h and the total RNA was purified with the Nucleospin RNA Clean-Up kit (Macherey-Nagel). The RNA concentrations were quantified using the Nanodrop 2000 spectrophotometer (Thermo Scientific, Illkirch, France). The integrity of the RNAs was verified on agarose gel.

### 4.5. Determination of the Level of Expression of Genes of Various Biosynthetic Pathways by RT-PCR

RT PCRs were carried out with 1 μg of total RNA per reaction. The absence of DNA in the samples was systematically checked by carrying out a control reaction without a reverse transcription (RT) step before the PCR step. The sequence of the primers used in this study are listed in Appendix A. Each pair of primers was tested beforehand in PCR on DNA. The RT PCR reactions were all carried out in a total volume of 50 μL, after the following cycle conditions: 50 °C for 30 min (reverse transcription), 97 °C for 15 min (activation of Taq polymerase) then 25 to 30 PCR cycles: 97 °C for 30 s (denaturation), 50 °C for 30 s (hybridization) and 72 °C for 30 s (elongation). The reaction products were analyzed by depositing 10 μL of reaction on a 1% agarose gel.

### 4.6. Assay of CDA, RED and ACT Production

To assay RED and ACT production, 10^6^ spores of *S. lividans*/pOSV206_*atpAGD* and *S. lividans*/pOSV206_*atpAGD*^Δ^ were plated, in triplicate, on a cellophane disk. This was laid down on the surface of 5 cm diameter plates of a solid HT medium, containing apramycin at 30 μg∙mL^−1^ and on a R2YE medium with no KH_2_PO_4_ added but containing 1 mM free K_2_HPO_4_ from elements of the media (condition of phosphate limitation) and containing apramycin at 50 μg∙mL^−1^, respectively. The plates were incubated at 28 °C. At different time points during the incubation period, mycelium was scraped off the cellophane disks of each replicate, lyophilized, weighted and used to RED that remained linked to the biomass as well as intracellular ACT.

RED production was assayed as described in [20]. To do so, 50 mg of dry mycelium was incubated in 1 mL of methanol for 5 min and then 1 mL of hydrochloric acid (1N) was added. The mixture was centrifuged for 10 mn at 10,000 *g*, and the optical density of the supernatant was measured at λ = 530 nm in a Shimadzu UV-1800 spectrophotometer. The concentration of RED in the extracts was calculated using the molar extinction coefficient of RED at 530 nm = 100,500 L∙mol^−1^∙cm^−1^, and the concentration of RED was expressed in nano moles of RED per mg of dry mycelium (Figure 4A).

Extra and intra-cellular ACT were also quantified as described in [20]. To quantify intracellular ACT, the ACT was extracted from the 50 mg of dry biomass upon incubation of the latter in 2 mL of 1 M KOH during 2 h at 4 °C under constant agitation. This operation was renewed once. After centrifugation, the supernatant was mixed with HCl 4 M (1 M final concentration) leading to precipitation of the ACT. The precipitated ACT pellet was re-suspended in 1 M KOH.

To quantify the extracellular ACT, the cellophane disk was lifted and the totality of the agar medium was taken from each replicate, sheared and allowed to diffuse in 20 mL of water for 2 h at 4 °C. The first eluate was transferred into a new tube, and 20 mL of water was added again to the agar medium and allowed to diffuse for 2 h at 4 °C. The second eluate was pooled with the first eluate, and 20 mL of water was added again to the agar medium and allowed to diffuse for 2 h at 4 °C. The last eluate was pooled to the other two and 3 mL of HCl (3 M) was added to 60 mL of the final eluate. The mixture was incubated on ice for 10 min to allow ACT precipitation. The precipitated ACT was collected by centrifugation (13,000× *g* for 10 min). Supernatants were discarded and the ACT pellets were re-suspended in 1 mL of KOH 1 M. Extra and intra- cellular concentration of the ACT was measured as optical density at 640 nm in a Shimadzu UV-1800 spectrophotometer using KOH 1 M as blank. The concentration of the ACT in the extracts was calculated using the molar extinction coefficient of ACT at 640 nm = 25,320 L∙mol^−1^∙cm^−1^ and the concentration of the ACT was expressed in nano moles of ACT per mg of dry mycelium (Figure 5).

To assay CDA production, the strains were grown, in triplicate, on a ONA medium in the presence or absence of calcium, and the CDA was assessed by the *Micrococcus luteus* inhibition bioassay (Figure 4B) as described in [9].

### 4.7. Determination of Total Lipid Content Using Attenuated Total Reflectance-Fourier Transform Infra Red Spectroscopy (ATR-FTIRS) Measurements

The total lipid content of *S.lividans*/pOSV206_*atpAGD* and *S.lividans*/pOSV206_*atpAGD*^Δ^ was determined from cultures grown, in triplicate, on a solid R2YE medium (apramycin at 50 μg∙mL^−1^) with no KH_2_PO_4_ added (condition of phosphate limitation) for 72 h as described in [23]. To do so, lyophilized mycelial samples were subjected to FTIR spectroscopy using a Bruker Vertex 70 FTIR spectrophotometer with a diamond ATR attachment (PIKE MIRacle crystal plate diamond ZnSe) and a MCT detector with a liquid nitrogen cooling system. A reference spectrum resulting from 100 averaged scans obtained in absence of any sample on the infrared support was acquired before each sample analysis. Scans were conducted from 3600 cm^−1^ to 600 cm^−1^ with a spectral resolution of 4 cm^−1^, with 100 averaged scans for each sample. This technique allows the establishment of spectral fingerprints of the complex biological structures under investigation. The C-H stretching bands of the CH_2_ of fatty acid chains (between 2959 and 2852 cm^−1^) and the C=O ester stretching band of the ester carbonyl (band near 1740 cm^−1^) are characteristic of lipids, including mainly polar membrane lipids and neutral lipids such as TAG. The height of the sharp and distinct C=O ester stretching band of the ester carbonyl is especially relevant for monitoring the total intracellular lipid contents of the strains [24]. Furthermore, since the biomass protein content can be directly characterized by the amplitude of the Amide I absorption band (1650 cm^−1^), all the FTIR spectra can be normalized to this band, allowing the comparison of the total lipid contents of the mycelial lawns of different strains. The total lipid contents of the strains assessed by FTIRS, expressed as arbitrary units, was converted into μg of Fatty Methyl Esters (FAME) per mg of dry mycelium, with the converting equation established by Millan-Oropeza et al. (2017) [23].

## 5. Conclusions

In conclusion, our conception of the relationships between a lowering of the intracellular ATP content and the triggering of antibiotics biosynthesis is summarized in the graphical representation shown in Figure 7. The lowering of the intracellular ATP content could be linked to a limitation in phosphate or to any situation leading to an ATP drain (stresses for instance). Consequently, any strategy leading to an artificial ATP depletion can constitute an interesting biotechnological tool to enhance the expression of the numerous cryptic pathways present in the *Streptomyces* genomes, and lead to the discovery of novel bioactive molecules potentially useful to human health. 

## Figures and Tables

**Figure 1 antibiotics-11-01157-f001:**
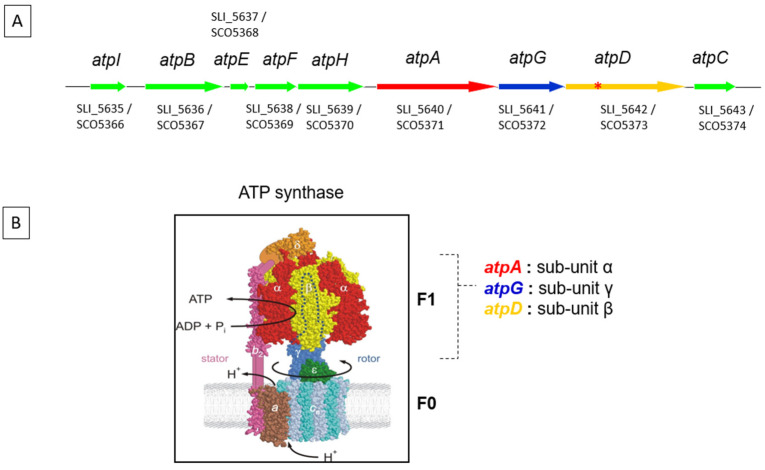
(**A**) Cluster of genes directing the synthesis of the ATP synthase sub-units of *Streptomyces lividans*. The red asterisk (*) in *atpD* indicates the 12 amino acids deletion of the active site of this sub-unit. (**B**) Schematic representation of the F0 and F1 part of the ATP synthase of *S. lividans.* When uncoupled from the membranous F0 part, the F1 part of the ATP synthase catalyzes the degradation of ATP (ATPase activity).

**Figure 2 antibiotics-11-01157-f002:**
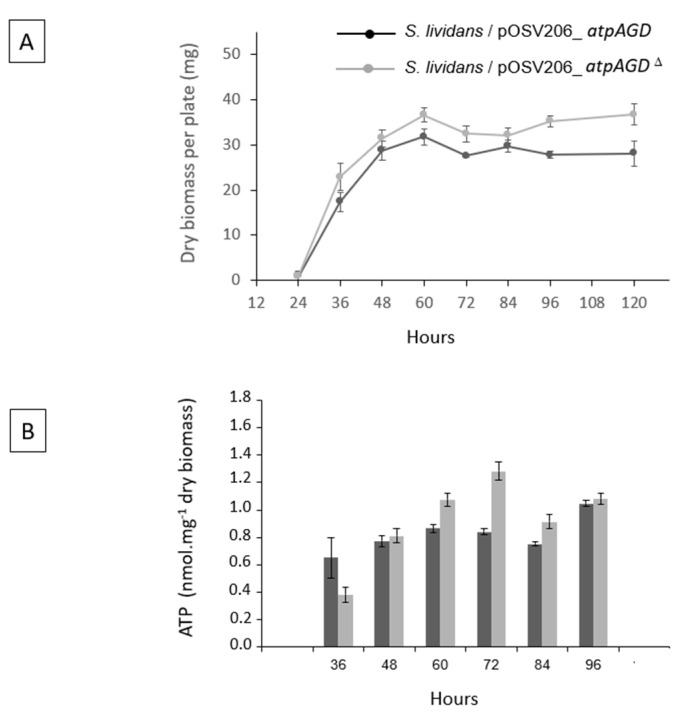
Growth curves of *S. lividans*/pOSV206_*atpAGD* (black line) and *S. lividans*/pOSV206_*atpAGD*^Δ^ (grey line) grown on solid HT medium (**A**) Assay of intracellular ATP concentration in *S. lividans*/pOSV206_*atpAGD* (dark grey histograms) and *S. lividans*/pOSV206_*atpAGD*^Δ^ (light grey histograms) at different points throughout growth (**B**).

**Figure 3 antibiotics-11-01157-f003:**
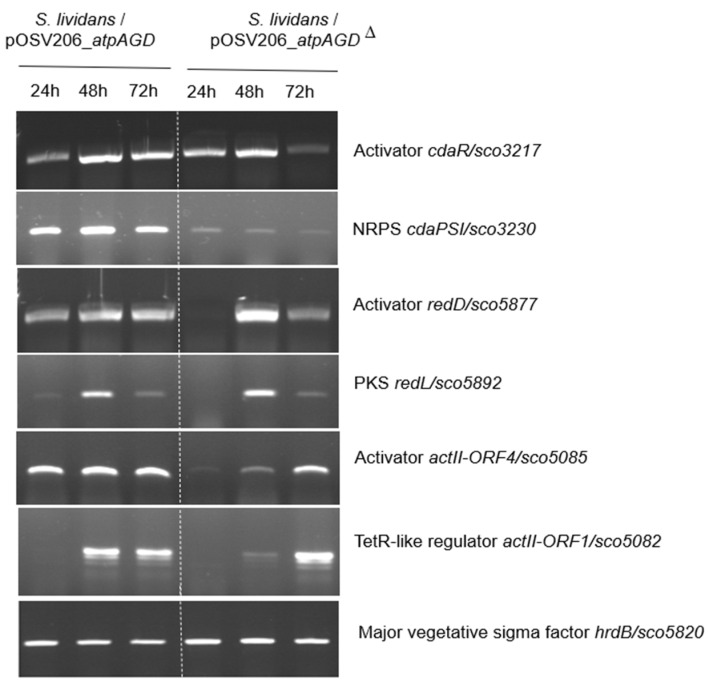
Determination by RT-PCR of the level of expression of the genes encoding the activator CdaR/SCO3217 and the type I peptide synthase SCO3230 of the CDA cluster; the activator RedD/SCO5877 and the polyketide synthase RedL/SCO5892 of the RED cluster, and of the activator ActII-ORF4/SCO5085 and the regulator ActII-ORF1/SCO5082 of the ACT cluster as well as HrdB/SCO5820, as internal control, using the primers specific of each gene listed in Appendix A. RNA was extracted from in *S. lividans*/pOSV206_*atpAGD* and *S. lividans*/pOSV206_*atpAGD*^Δ^ grown on solid HT medium for 48 h.

**Figure 4 antibiotics-11-01157-f004:**
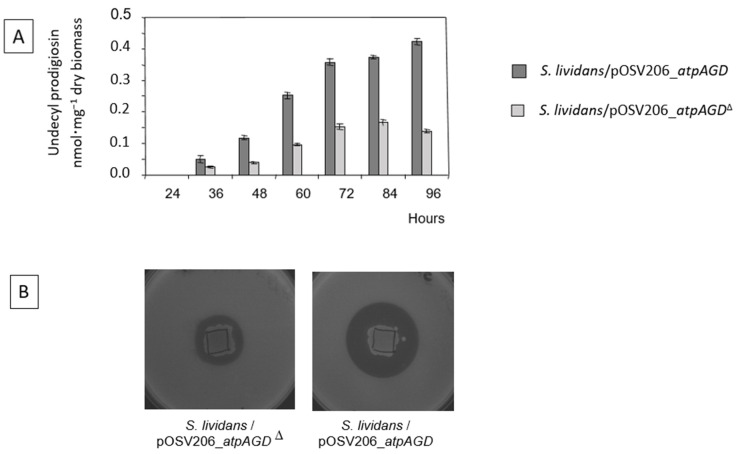
Quantification of undecyprodigiosin (RED) production throughout growth of *S. lividans*/pOSV206_*atpAGD* (dark grey histograms) and *S. lividans*/pOSV206_*atpAGD*^Δ^ (light grey histograms) grown for 48 h on solid HT medium (**A**) and detection of CDA production via the formation of halos of inhibition of *Micrococcus luteus* growth around the deposit of a piece of agar culture of *S. lividans*/pOSV206_*atpAGD* and *S. lividans*/pOSV206_*atpAGD*^Δ^ grown on ONA medium for 48 h in presence of calcium (**B**).

**Figure 5 antibiotics-11-01157-f005:**
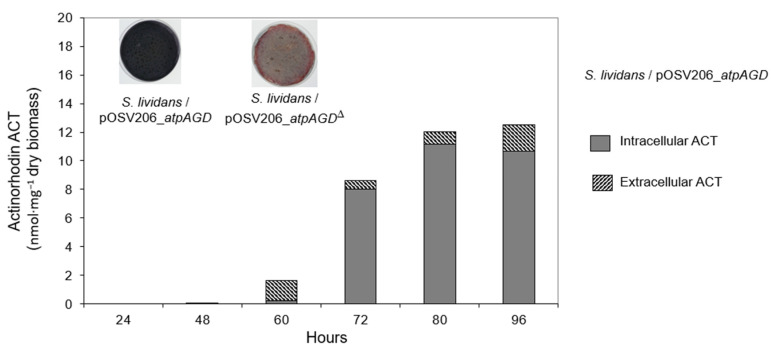
Quantification of intra (dark grey part) and extra (hatched part) cellular production of ACT throughout growth of *S. lividans*/pOSV206_*atpAGD* grown on solid R2YE medium limited in Pi for 48 h. The production of ACT by *S. lividans*/pOSV206_*atpAGD*^Δ^ is not shown since it was below detection limits.

**Figure 6 antibiotics-11-01157-f006:**
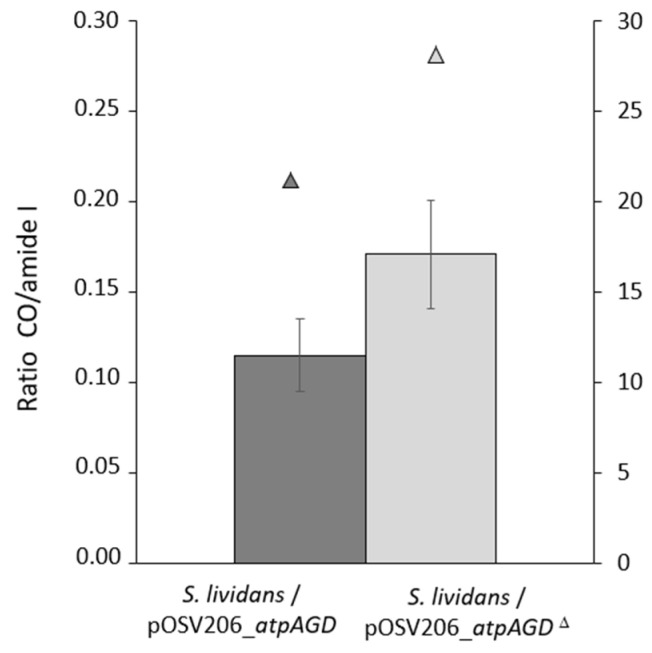
Fourier Transform Infra Red Spectroscopy (FTIRS) analysis of total lipid content of *S. lividans*/pOSV206_*atpAGD* (dark grey histograms) and *S. lividans*/pOSV206_*atpAGD*^Δ^ (light grey histograms) grown at 28 °C for 72 h on R2YE with no K_2_HPO_4_ added. Triangles above the histograms represent the biomass of the strains expressed as mg of dry mycelium per plate.

**Figure 7 antibiotics-11-01157-f007:**
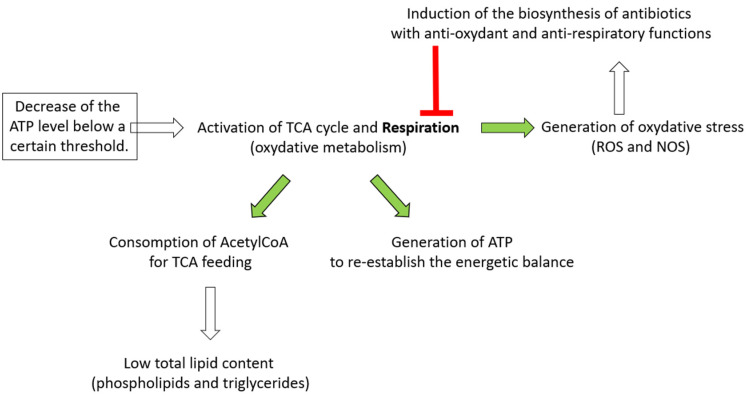
Schematic representation of the link between the lowering of the intracellular ATP concentration, the triggering of the biosynthesis of a specific class of bioactive specialized molecules and the negative role played by the latter in regulation of the respiratory activity of the producing strain.

## Data Availability

Data is contained in this article or supplementary material.

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
