# Peer review of "The Generation of an Artificial ATP Deficit Triggers Antibiotic Production in Streptomyces lividans"

_antibiotics, 2022, doi:10.3390/antibiotics11091157_

Round 1

Reviewer 1 Report

review: "The generation of an artificial ATP deficit triggers antibiotic production in Streptomyces lividans"

The present article deals with insufficient phosphate levels, which is associated with a low intracellular ATP content compared to growth when phosphate levels are adequate; most Streptomyces species start producing antibiotics. This finding revealed that low ATP levels might immediately trigger antibiotic biosynthesis. The topic is very interesting, and the article is well written, although I would suggest some minor changes.

First, I would like to point out that the abstract seems to be a little bit lengthy. Additionally, I would add a few more bibliographic positions to the introduction section and highlight this work's novelty.

I would also suggest moving the materials and method section after the introduction. So the reader should know what was the experimental procedure and then read the results.  Please also mention how many times did you conduct each experiment.

Author Response

Reviewer 1

The present article deals with insufficient phosphate levels, which is associated with a low intracellular ATP content compared to growth when phosphate levels are adequate; most Streptomyces species start producing antibiotics. This finding revealed that low ATP levels might immediately trigger antibiotic biosynthesis. The topic is very interesting, and the article is well written, although I would suggest some minor changes.

First, I would like to point out that the abstract seems to be a little bit lengthy.

The abstract was shortened and modified.

Additionally, I would add a few more bibliographic positions to the introduction section and highlight this work's novelty.

We thank the reviewer for his/her positive comments and have now modified the text to highlight in a better way the novelty of our work.
That is true that some references are missing, unfortunately I am on holidays and I forgot to take my EndNote file with me so I will not able to include the necessary references before end of august.  

I would also suggest moving the materials and method section after the introduction. So the reader should know what was the experimental procedure and then read the results.

I think that the layout is imposed by the journal.

  Please also mention how many times did you conduct each experiment.

Error bars indicate that experiments were carried out in triplicates, so we have now made this statement more clear in materials and methods.

Reviewer 2 Report

English grammar and spellings to be checked thoroughly. Figures levels needs to be used uniform font and style (Figure 5 & 6). Outcome of the study need to be addressed properly.

Author Response

Reviewer 2

English grammar and spellings to be checked thoroughly.

We apologize about this problem, english grammar and spellings have now been checked and corrected throughout the text.

Figures levels needs to be used uniform font and style (Figure 5 & 6).

We thank the reviewer for this comment, arial 14 is now used in all the figures.

Outcome of the study need to be addressed properly.

We did our best to improve the discussion in the revised version of the manuscript.

Reviewer 3 Report

The manuscript entitled ‘The generation of an artificial ATP deficit triggers antibiotic production in Streptomyces lividans’ authored by Seghezzi et al discusses how low ATP levels cause the model strain, Streptomyces lividans to produce more antibiotics. The expression of a functioning ATPase reduces the amounts of ATP. As a control, a mutant ATPase has also been overexpressed. The paper discusses a compelling premise. The writing and experimental strategy, though, could have been enhanced.

Include statements in the opening highlighting the novelty of the work or reference works that have done similar work.

Discuss the studies involved in CDA, ACT, and RED production in Streptomyces lividans/ Streptomyces coelicolor

Discuss briefly the rationale behind selecting Streptomyces lividans for this study

Why is the ATP concentration in Fig. 2 not measured in a phosphate-limited environment? If it has already been reported, discuss the studies that have established this hypothesis.

Although the production levels of RED activators are low, their expression profiles are similar. Explain the difference.

Discuss whether the higher ACT was due to ATPase activity alone or a phosphate shortage in conjunction with ATPase.

Describe how a slight ATP deficiency led to a significant increase in the ACT. Whether oxidative stress is what caused it. Have you tried to confirm it?

Discuss the differential expression of the activators

The discussion section of the manuscript needs to be more elaborate

Why aren't growth and ATP synthesis tests performed on R2YE medium and oxoid nutrition agar?

Growth experiments and ATP measurements were carried out in the HT medium but the lipid content analysis was carried out in the R2YE medium. Why?

Why the lipid content is measured only at the end of 72h? Including two more time points would have added more value to the experiment.

The authors might suggest the future directions and/or research gap of antibiotic synthesis in S. lividans at the end of the manuscript

Minor comments:

The manuscript must be thoroughly revised and proofread for grammatical errors and typos.

The antibiotic name ‘Calcium Dependant Antibiotic (CDA)’ is written in capital letters while the other antibiotics are written in small letters. The authors should maintain consistency throughout the manuscript.

Abbreviations of the antibiotics may be used instead of the full names everywhere.

In some places, ONA is used and in some places Oxoid Nutrient Medium is used. Use the full form only the first time.

The authors are suggested to replace ‘4.1. Construction of the plasmids used in this study’ with ‘4.1. Plasmid construction’

In Fig. 1 synthesis of the ATP synthase sub-units

In Fig. 2B 24 h missing?

In Fig. 2A, 2B and 4A – x-axis – Time (hours) and fig.2A y-axis units?

Author Response

Reviewer 3:

The manuscript entitled ‘The generation of an artificial ATP deficit triggers antibiotic production in Streptomyces lividans’ authored by Seghezzi et al discusses how low ATP levels cause the model strain, Streptomyces lividans to produce more antibiotics. The expression of a functioning ATPase reduces the amounts of ATP. As a control, a mutant ATPase has also been overexpressed. The paper discusses a compelling premise. The writing and experimental strategy, though, could have been enhanced.

Include statements in the opening highlighting the novelty of the work or reference works that have done similar work.

We have now modified the text to highlight in a better way the novelty of our work.
To our knowledge, we are the only publishing group on this line of research that is why we are forced to self-citation.

Discuss the studies involved in CDA, ACT, and RED production in Streptomyces lividans/ Streptomyces coelicolor

It is just important to be aware that these pathways are present and functional in both strains but are mainly strongly expressed in S. coelicolor.

We apologize that some references concerning CDA, RED and ACT as well as other references are obviously missing in the introduction but unfortunately I am on holidays and I forgot to take my EndNote file with me so I will not be able to include the necessary references before end of august.

Discuss briefly the rationale behind selecting Streptomyces lividans for this study.

We have now modified the introduction to  explain more clearly why S. lividans was choosen for this study. S. lividans was choosen as it is a weak antibiotic producing strain and we expected to enhance its weak antibiotic producing ability using the strategy described in the manuscript.

Why is the ATP concentration in Fig. 2 not measured in a phosphate-limited environment?
If it has already been reported, discuss the studies that have established this hypothesis.

HT is a medium limited in phosphate, no K2HPO4 was added to this medium.
To our knowledge, the only other study where ATP was assayed in S. lividans and S. coelicolor grown in another medium, R2YE limited or proficient in phosphate, is originating from our group.

[5] C. Esnault, T. Dulermo, A. Smirnov, A. Askora, M. David, A. Deniset-Besseau, I.B. Holland, and M.J. Virolle, Strong antibiotic production is correlated with highly active oxidative metabolism in Streptomyces coelicolor M145. Sci Rep 7 (2017) 200.

Although the production levels of RED activators are low, their expression profiles are similar. Explain the difference.

We have no real satisfactory explanation for these differences, we can only say that S. lividans spontaneously produces RED more readily than ACT.

Discuss whether the higher ACT was due to ATPase activity alone or a phosphate shortage in conjunction with ATPase.

That is probably the conjonction of both processes that leads to an enhanced ACT production. Phosphate limitation leads to a low ATP content further decreased below a certain threshold by the ATPase activity.
As for the ppk (ref 5) or pptA (ref 7) mutants of S. lividans that are over-producing ACT, the addition of phosphate will most probably inhibits ACT production in the strain over-expressing the ATPase.

Describe how a slight ATP deficiency led to a significant increase in the ACT.
Whether oxidative stress is what caused it. Have you tried to confirm it?

Our conception is summarized in Figure 7 of the revised version of the manuscript, in the discussion and in refs 5, 10 and 12.

An ATP deficiency leads to an activation of the oxidative metabolism that generates ATP, in order to re-establish the cellular energetic balance. However a strong activation of the oxidative metabolism leads to the generation of ROS/RNS (oxidative stress) that would trigger the expression of bio-active molecules combating oxidative stress and reducing the efficiency of respiration to limit oxidative stress as well as ATP generation, especially in a context of limited phosphate availability. The production of these bio-active molecules constitutes somehow a negative regulatory feedback loop to “cool down” respiration and thus limit ATP generation as well as detrimental effects linked to the generation of ROS/NOS.

For instance, S. coelicolor is characterized by a 2 to 3 fold higher ATP content than S. lividans and this implies an highly active oxidative metabolism of this strain. Interestingly, in S. coelicolor, the onset of ACT production coincides with an abrupt drop in the ATP content of this strain suggesting an anti-respiratory function of ACT (ref 5). We have also demonstrated that ACT bears an anti-oxydant function (supplementary data of ref 13) and we are preparing a paper demonstrating that the addition of some specific anti-oxidant molecules in the growth medium of S. coelicolor abolishes ACT production.

Discuss the differential expression of the activators

We just think that the expression of the activators is induced by oxidative stress but details of the regulatory cascade are missing.

The discussion section of the manuscript needs to be more elaborate

We did our best to improve the discussion in the revised version of the manuscript.

Why aren't growth and ATP synthesis tests performed on R2YE medium and oxoid nutrition agar?

The assay of the intracellular ATP concentration is extremely tricky and time consuming that is why we have only done it in HT medium.

Growth experiments and ATP measurements were carried out in the HT medium but the lipid content analysis was carried out in the R2YE medium. Why?

This comment is indeed relevant. We carried out this study in R2YE because that is in this medium that the polyketide antibiotic ACT is more abundantly produced and also because we published many studies concerning the link between the lipid content of various Streptomyces strains and their antibiotic producing abilities (refs 5, 7, 8, 9) and in these studies the strains were grown in the classical R2YE medium.

Why the lipid content is measured only at the end of 72h? Including two more time points would have added more value to the experiment.

In the course of our study published as ref 9, we assayed lipids a different time points (36, 48, 60, 72, 84, 96 h) and noticed that the lipid content varies a lot throughout growth but stabilizes at 72h, that is why we assayed lipids at this time point.

The authors might suggest the future directions and/or research gap of antibiotic synthesis in S. lividans at the end of the manuscript.

We did our best to improve the discussion in the revised version of the manuscript.

Minor comments:

The manuscript must be thoroughly revised and proofread for grammatical errors and typos.

This was done, we apologize about this errors.

The antibiotic name ‘Calcium Dependant Antibiotic (CDA)’ is written in capital letters while the other antibiotics are written in small letters. The authors should maintain consistency throughout the manuscript.

This was corrected.

Abbreviations of the antibiotics may be used instead of the full names everywhere.

This was corrected.

In some places, ONA is used and in some places Oxoid Nutrient Medium is used. Use the full form only the first time.

This was corrected.

The authors are suggested to replace ‘4.1. Construction of the plasmids used in this study’ with ‘4.1. Plasmid construction’

This was changed.

In Fig. 1 synthesis of the ATP synthase sub-units

This was corrected.

In Fig. 2B 24 h missing?

Yes because at 24h the amount of collected biomass in one plate is too low to assay ATP reliably.

In Fig. 2A, 2B and 4A – x-axis – Time (hours) and fig.2A y-axis units?

We are very grateful to the reviewer to pay attention to details of our figures and we made the necessary modifications.

Round 2

Reviewer 3 Report

The authors have addressed the comments and revised the manuscript appropriately. The manuscript is recommended for publication in its present form.